# Perspectives of nurses' role in interprofessional pharmaceutical care across 14 European countries: A qualitative study in pharmacists, physicians and nurses

Elyne De Baetselier[1]*, Tinne Dilles[1], Luis M. Batalha[2], Nienke E. Dijkstra[3], Maria I. Fernandes[2], Izabela Filov[4], Juliane Friedrichs[5], Vigdis A. Grondahl[6], Jana Heczkova[7], Ann Karin Helgesen[6], Sue Jordan[8], Sarah Keeley[9], Thomas Klatt[5], Petros Kolovos[10], Veronika Kulirova[7], Sabina Ličen[11], Manuel Lillo-Crespo[12], Alba Malara[13], Hana Padysakova[14], Mirko Prosen[11], Dorina Pusztai[15], Jorge Riquelme-Galindo[12], Jana Rottkova[14], Carolien G. Sino[3], Francesco Talarico[13], Styliani Tziaferi[10], Bart Van Rompaey[1]

1 Faculty of Medicine and Health Sciences, University of Antwerp, Antwerp, Belgium, 2 Health Sciences Research Unit: Nursing (UICISA: E), Nursing School of Coimbra (ESEnfC), Coimbra, Portugal, 3 Research Group Care for the Chronically Ill, University of Applied Sciences Utrecht, Utrecht, The Netherlands, 4 University "St. Kliment Ohridski" Bitola, Bitola, Republic of North-Macedonia, 5 Medical Faculty, Institute of Health and Nursing Science, Martin Luther University Halle-Wittenberg, Halle (Saale), Germany, 6 Faculty of Health and Welfare, Østfold University College, Halden, Norway, 7 Institute of Nursing Theory and Practice, First Faculty of Medicine, Charles University, Prague, Czech Republic, 8 Department of Nursing, Swansea University, Swansea, Wales, United Kingdom, 9 Department of Nursing and Clinical Science, Bournemouth University, Bournemouth, England, United Kingdom, 10 Department of Nursing, Laboratory of Integrated Health Care, University of Peloponnese, Sparti, Greece, 11 Faculty of Health Sciences, Department of Nursing, University of Primorska, Izola, Slovenia, 12 Department of Nursing, Faculty of Health Sciences, University of Alicante, Alicante, Spain, 13 ANASTE-Humanitas Foundation, Rome, Italy, 14 Faculty of Nursing and Professional Health Studies, Slovak Medical University in Bratislava, Bratislava, Slovak Republic, 15 Institute of Nursing Sciences, Basic Health Sciences and Health Visiting, Faculty of Health Sciences, University of Pécs, Pécs, Hungary

* Elyne.DeBaetselier@uantwerpen.be

**Data Availability Statement:** Ethical restrictions have been imposed on data sharing by the Ethics Committee for Social Sciences and Humanities of

## Abstract

### Objectives

To understand healthcare professionals' experiences and perceptions of nurses' potential or ideal roles in pharmaceutical care (PC).

### Design

Qualitative study conducted through semi-structured in-depth interviews.

### Setting

Between December 2018 and October 2019, interviews were conducted with healthcare professionals of 14 European countries in four healthcare settings: hospitals, community care, mental health and long-term residential care.

the University of Antwerp, the Medical Ethics Committee of the Republic of Slovenia and the UK NHS Research Ethics Committee that approved this study. The data contain potentially identifying and sensitive information. Also, investigators from the other countries confirmed that making these sensitive data publicly available without having requested the consent of the interviewees beforehand, is impossible for ethical and legal concerns. UK data are stored at Swansea University, Swansea, UK. All proposals to view the data are subject to review by Swansea University's Research Governance department and the PI. Before any data can be accessed, approval must be given. The application process is via the Academic Lead for Research Integrity Research Engagement & Innovation Services, Swansea University and the PI or Neil Carter. Contacts: Swansea University, Swansea SA2 8PP Tel: +44 /0 1792 606060 and 518541 or 295610 Email: researchgovernance@swansea.ac.uk, s.e. jordan@swansea.ac.uk or n.carter@swansea.ac.uk Data from the other 13 countries are stored at the University of Antwerp, Antwerp, Belgium. All proposals to view the data are subject to review by University of Antwerp's Research Governance department and the PI. Before any data can be accessed, approval must be given. The application process is via the data protection officer, Antwerp University and the PI. Contacts: University of Antwerp, Prinsstraat 13, 2000 Antwerp, Belgium. Email: privacy@uantwerpen.be, Tinne. Dilles@uantwerpen.be.

**Funding:** The research was supported by the Erasmus+ Programme of the European Union [grant number 2018-1-BE02-KA203-046861] and MDMJ accountants, an accountancy service in Belgium that financially supported the Belgian authors (www.mdmj.be). The funders had no role in study design, data collection and analysis, decision to publish, or preparation of the manuscript.

**Competing interests:** The study "Perspectives of nurses' roles in interprofessional pharmaceutical care across 14 European countries: a qualitative study in pharmacists, physicians and nurses" was supported by the Erasmus+ Programme of the European Union [grant number 2018-1-BE02-KA203-046861] and MDMJ accountants, Belgium (https://www.mdmj.be). The funders had no role in study design, data collection and analysis, decision to publish, or preparation of the manuscript. Within this Competing Interests Statement, we explicitly elaborate on MDMJ accountants as a commercial co-funder of this study. This organisation is an accountancy service in Belgium that financially

## Participants

In each country, pharmacists, physicians and nurses in each of the four settings were interviewed. Participants were selected on the basis that they were key informants with broad knowledge and experience of PC.

## Data collection and analysis

All interviews were conducted face to face. Each country conducted an initial thematic analysis. Consensus was reached through a face-to-face discussion of all 14 national leads.

## Results

340 interviews were completed. Several tasks were described within four potential nursing responsibilities, that came up as the analysis themes, being: 1) monitoring therapeutic/ adverse effects of medicines, 2) monitoring medicines adherence, 3) decision making on medicines, including prescribing 4) providing patient education/information. Nurses' autonomy varied across Europe, from none to limited to a few tasks and emergencies to a broad range of tasks and responsibilities. Intended level of autonomy depended on medicine types and level of education. Some changes are needed before nursing roles can be optimised and implemented in practice. Lack of time, shortage of nurses, absence of legal frameworks and limited education and knowledge are main threats to European nurses actualising their ideal role in PC.

## Conclusions

European nurses have an active role in PC. Respondents reported positive impacts on care quality and patient outcomes when nurses assumed PC responsibilities. Healthcare professionals expect nurses to report observations and assessments. This key patient information should be shared and addressed by the interprofessional team. The study evidences the need of a unique and consensus-based PC framework across Europe.

## Introduction

Effective team communication and clear definitions of roles are two of the fundamental prerequisites for effective collaboration among nurses, physicians and pharmacists to deliver high quality care and better meet patients' needs [1, 2]. Unclear role boundaries hinder collaboration on different levels: quality of interprofessional communication and collaboration in daily clinical practice; transnational collaboration in research, education and innovation; and labor mobility of nurses [1–4]. A clear description of roles in pharmaceutical care (PC) and medicines optimisation, however, is not always available [2, 5–7]. In this study PC is defined as *'Healthcare professionals' contribution to the care of individuals in order to optimize medicines use and improve health outcomes'*. This definition is based on the definition of the Pharmaceutical Care Network Europe (PCNE) [8], which, however, was limited to the contribution of pharmacists, as well as the original definition of Hepler and Strand in 1990 [9]. After all, the need for interprofessional collaboration in PC is broadly recognised [3, 10–14].

Large variations in nurses' roles exist, as was demonstrated in a cross-country comparative study in 39 countries. In two third of the countries, nurses took up advanced roles from

supported the Belgian authors. No competing interests interfered with, or could reasonably be perceived as interfering with, the full and objective presentation of this research. Next to the financial support, no professional or personal competing interests can be declared about this commercial funder. The financial support consisted of a gift of 27000€, to be used as co-funding of the Belgian research team, next to the funding of the Erasmus+ programme of the European Union. No other statements relating to employment, consultancy, patents, products in development or marketed products can be declared. The Belgian authors confirm that the financial support of MDMJ accountants does not alter our adherence to PLOS ONE policies on sharing data and materials.

physicians, but the extent varied. A trend towards expanding nurses' scope-of-practice in primary care was evolving [4]. The large variation in nurses' roles was corroborated in the EUPRON-study investigating nurses' current clinical practices in interprofessional pharmaceutical care (PC). This showed that monitoring medicines effects, monitoring medicines adherence, prescribing medicines and providing patient education/information about medicines are already part of nurses' clinical practice, and nurses' contribution to PC differs between countries, in both law and practice [13].

Nurses' scope of practice is considered as the full range of roles, responsibilities and tasks that nurses are educated, competent and authorized to perform [15]. Within this scope of practice, a framework for nurses' ideal roles in interprofessional PC would allow insights into current and potential roles in PC, and facilitate discussions in clinical practice, education, research, international comparisons, policy-making and legislation. Additionally, this framework could be used to develop an assessment to evaluate nurse competences in PC, as a guidance to evaluate nurse education, as a tool for nurse educators, for benchmarking and nurse labour mobility. To date, we have not identified such a framework in the published literature. To develop a robust framework, adapted to the needs of clinical practice, insights in the preferences of the most important stakeholders (nurses, physicians and pharmacists) are necessary. Exploring those preferences, requires in-depth qualitative research.

This study is the second part of the DeMoPhaC project, an international Erasmus + collaboration to investigate nurses' role in interprofessional PC in 14 countries. Within this project several large-scale quantitative and qualitative studies are being undertaken with healthcare workers and nursing students. The overall aim of the project is the Development of a Model for nurses' role in interprofessional Pharmaceutical Care in Europe and the development of an assessment to evaluate nursing curricula and final year nursing students' competences in PC. The first part of the project focused on the current clinical practice of nurses in PC without insights into strenghts, weaknesses, opportunities and threats from nurses' involvement in PC [13]. In-depth qualitative research through case studies can close this gap. Therefore, we aimed to perform a qualitative study, to understand pharmacists', physicians' and nurses' experiences and perceptions of nurses' potential or ideal roles in PC.

By considering the 'potential or ideal roles', we aimed to investigate nurses' responsibilities and tasks within–but also beyond–nurses' current legal scope of practice, taking into account all necessary contextual factors.

## Methods

### Study design

This study was conducted and reported according to the Consolidated Criteria for Reporting Qualitative Research (COREQ) [16].

We explored nurses', physicians' and pharmacists' expectations about nurses' role in PC, and related strengths, weaknesses, opportunities and threats through a qualitative descriptive research design with a phenomenological case study approach. Case study as a research method has been widely used for preliminary and exploratory stages of research [17–19]. Multiple case studies allow cross-case comparisons and the identification of themes across cases. A phenomenological approach using in-depth semi-structured interviews within the case studies support high quality data collection [20–22]. Phenomenology is well suited for exploring perspectives of healthcare professionals [23]. This research approach was chosen as an appropriate way to describe the essence of the phenomenon "nurses' role in interprofessional PC", by exploring it from the perspective of those who have experienced it, namely pharmacists, physicians and nurses themselves. Interviewing this study population enables studying and

understanding healthcare professionals' lived experiences in interprofessional PC. Only by understanding their personal experiences and perceptions of nurses' responsibilities and tasks, and interprofessional collaboration and communication, we will be able to provide detailed examination of the current strengths and weaknesses, together with the future opportunities and threats from nurses' involvement in PC [23].

## Setting

The study took place in 14 European countries: Belgium, Czech Republic, Germany, Greece, Hungary, Italy, the Republic of North Macedonia, the Netherlands, Norway, Portugal, Slovakia, Slovenia, Spain and the United Kingdom (England and Wales). In each country in-depth interviews were conducted in four different settings: hospitals, community care, long-term residential care, and mental health care.

## Participants

'Key informant' pharmacists, physicians and nurses were purposively sampled [24]. They could only be selected on the condition that they were named as expert in PC by at least two other healthcare professionals, with local knowledge of PC, and insights into the nature of problems and possible solutions. This allowed us to get information about nurses' roles in interprofessional PC and to understand the motivations and beliefs of a large number of healthcare professionals with diverse backgrounds and opinions. Representatives of professional associations for nurses, physicians and pharmacists, and healthcare providers in different healthcare institutions were asked to identify key informants. Researchers contacted the persons identified as potential participants by email or telephone, informed them about the study, and about being named as a key informant on nurses' role in interprofessional PC. If they agreed with being able to serve as a key informant, written information was provided to fully inform the potential participants about the study details.

We aimed for at least two interviews per profession (n = 3) per healthcare setting (n = 4), per country (n = 14), resulting in 24 in-depth face-to-face interviews per country. These numbers were aimed for in order to compile a sample with perspectives as diverse as possible. Data saturation was reached in each participating country. There were no restrictions as to gender or age. No reimbursement was provided for participation. Exact numbers of those approached and declining were not registered in all countries.

## Interview guide development

An interview guide (S1 Appendix) was developed in English based on literature and the results of a previous quantitative study about nurses' practices in interprofessional PC (Fig 1, step 1) [13]. During a meeting with all European partners, the interview guide was adjusted until consensus was reached (Fig 1, step 2).

To ensure conformity across twelve languages, the concept of PC was described at the beginning of the interview: "*healthcare professionals' contribution to the care of individuals in order to optimize medicines use and improve health outcomes*". This description was derived from the Pharmaceutical Care Network Europe definition of 2013, taking into account the interprofessional aspect of PC [8, 14].

Responsibilities and tasks were defined based on the literature, together with discussions with an expert in health law, liability law and ethics and an expert in legal philosophy and ethics: "*The role of nurses involves several responsibilities. A responsibility for nurses is an obligation that they have by virtue of their role as a nurse. Their central responsibility is to be the patient's health advocate and to provide high quality care, using sound professional judgement and taking*

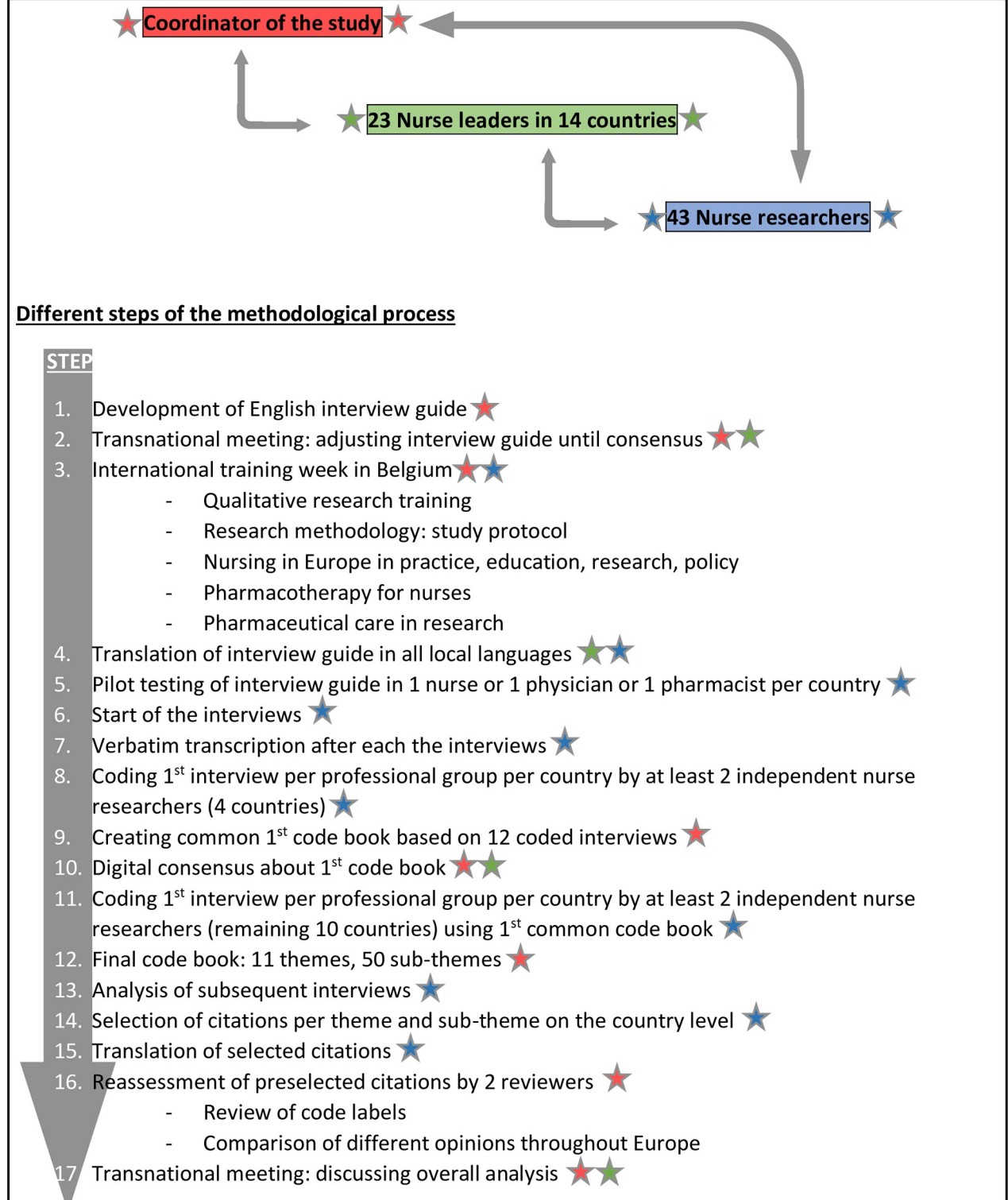

**Fig 1. International approach to increase methodological quality.**

*into account the relevant legal and moral considerations. The other responsibilities of nurses derive from this central responsibility. Nurses can be made to answer for failing in their responsibilities, which could result in disciplinary, civil, and criminal liability. Specific tasks may have to be performed in order to fulfill a responsibility.*" [25, 26].

The interview guide consisted of four main topics.

**Topic 1: Responsibilities**. Respondents were asked what responsibilities would be part of the ideal role of nurses in PC and what these responsibilities would imply. Preparation and administration of medicines by nurses was considered as an obvious part of PC and therefore outside the study's scope. After open reflections, four responsibilities were presented: 1) monitoring and following-up of therapeutic and adverse effects of medicines; 2) monitoring and following-up medicines adherence; 3) decision making on medicines use, including prescribing medicines, excluding preparation and administration; 4) providing patient education and information about medicines. Respondents were asked what they would like to change, add, or remove. This structuring ensured uniformity across 14 countries and 12 languages.

**Topic 2: Tasks**. Specific tasks within the previously defined responsibilities were elicited. A similar strategy as above, with open and then more guided reflections, was used. The predefined tasks for reflection were: 1) detecting clinical change, healthcare problems or assessing patient needs; 2) registration; 3) interprofessional communication (including reporting, alerting and discussion); 4) patient communication; 5) intervention in emergency cases; 6) follow-up; 7) self-care support; 8) 'dependent' nurse prescribing; 9) 'independent' nurse prescribing; 10) reporting medication errors and safety issues.

**Topic 3: Interprofessional team working**. Ideal communication and collaboration between pharmacists, physicians and nurses, when aiming for high quality PC and predefined interactions were suggested: 1) nurses reporting observations to physicians and pharmacists; 2) physicians providing information and instruction to nurses; 3) pharmacists giving advice to nurses.

**Topic 4: SWOT analysis**. Finally, respondents were asked to reflect on strengths, weaknesses, opportunities and threats (SWOT) of nurses' current and ideal roles.

## Data collection

Nurse researchers in each country were trained in qualitative research and in-depth interviewing during a joint one-week training program at University of Antwerp in November 2019 (Fig 1, step 3). When agreed, the interview guide was translated into all national languages and pilot tested in each country by at least one pharmacist, physician and nurse (Fig 1, step 4–5). The test interviews were not included in the data analysis. No significant adjustments were made after the pilot interviews. Between December 2018 and October 2019 interviews were conducted by two to four interviewers per country (Fig 1, step 6). Participants were mostly interviewed at their workplace, or another location, such as participant's home or the researcher's workplace. Regardless of location, confidentiality was maintained. Only the interviewer and the interviewee were present during the interview. Interviews lasted from 30 to 90 minutes, and were audio recorded. Field notes were taken. No interviews were repeated. Audio recordings were transcribed *verbatim* by the interviewer or a professional transcriber (Fig 1, step 7). They were not returned to participants for member checking.

## Data analysis

The qualitative analysis started after the first interview [27]. The transcripts were coded by labelling lines of text in order to group and compare similar or related data segments. To create an international code book for data analysis, 12 interviews were fully translated into English and coded by the local researchers from 4 countries (one pharmacist, one physician, one nurse per country) (Fig 1, step 8). The English codes were then collected to create a common first code book, to be used as a guide for analysing subsequent interviews (Fig 1, step 9). Consensus was achieved within the consortium, and the next 30 interviews were analysed (Fig 1, step 10–11). Extra codes and themes could be added if new content arose. The final code book consisted of 11 themes, combined with 49 sub-themes, addressing nurses' roles and the related SWOT analysis (S2 Appendix; Fig 1, step 12).

To improve the confirmability of the study, every first interview per professional group per country was analysed by two researchers [28–30]. In that way, at least three interviews per country (one nurse, one physician, one pharmacist) were analysed by two researchers. All other transcripts and coding were at least checked by a colleague. After the data were analysed at national level, by coding the transcripts, researchers in each country selected quotations for each theme and sub-theme (Fig 1, step 14). To store the quotations, add labels and arrange the data, Microsoft Excel® tables were created. To accomplish an overall view on the data, the pre-selected citations were reviewed by two researchers (first and second author) to reassess the code labels for accuracy and to compare the different opinions throughout Europe. All assumptions were taken into account, regardless the number of times they occurred (Fig 1, step 16). The national data per country, as well as the overall international data, were presented at an international meeting with all partners to discuss the completeness and interpretation of the results per country, and achieve international consensus (Fig 1, step 17).

## Ethics approval

The Ethics Committee for Social Sciences and Humanities of the University of Antwerp approved the study design (reference SHW_19_30). Depending on local regulations in Slovenia, the UK and Portugal, additional approval was obtained from the Medical Ethics Committee of the Republic of Slovenia (reference 0120-516/2018/6), Health and Care Research Wales (reference 19/HCRW/00) and the Ethics Committee of the Escola Superior de Enfermagem in Coimbra (reference 543/12-2018). National regulations and laws applying to the other countries didn't require additional permits or approvals. All respondents received information on the purpose, design and execution of the study. Written informed consent was given by all study participants.

## Results

The characteristics of the 340 healthcare professionals interviewed are presented in Table 1: 113 pharmacists, 111 physicians and 116 nurses, employed in hospital care (45%), community care (26%), residential care (14%), mental healthcare (9%), and other settings, such as a (10%). Healthcare professionals involved were equally distributed across participating countries. Most respondents worked in clinical practice (80%) and spent an estimated mean of 29 ± 15.1 hours/week on PC.

In response to questions about the ideal role of nurses in clinical practice, the four main responsibilities, developed in previous work, remained substantially unchanged. Within each responsibility, several tasks and contextual factors were reported. Opinions differed regarding expectations of nurses. An overview of all nurse responsibilities and tasks in interprofessional PC reported by pharmacists, physicians and nurses is given in Table 2.

**Table 1. Population characteristics.**

| | n (%) | |
|---|---|---|
| **Country** | | |
| Belgium | 28 (8.2) | |
| Czech Republic | 29 (8.5) | |
| Germany | 22 (6.5) | |
| Greece | 24 (7.1) | |
| Hungary | 21 (6.2) | |
| Italy | 24 (7.1) | |
| The Netherlands | 24 (7.1) | |
| Norway | 24 (7.1) | |
| Portugal | 24 (7.1) | |
| Republic of North Macedonia | 24 (7.1) | |
| Slovakia | 24 (7.1) | |
| Slovenia | 24 (7.1) | |
| Spain | 24 (7.1) | |
| United Kingdom | 24 (7.1) | |
| **Profession** | | |
| Pharmacist | 113 (33.2) | |
| Physician | 111 (32.6) | |
| Nurse | 116 (34.1) | |
| **Gender** | | |
| Female | 206 (60.6) | |
| Male | 134 (39.4) | |
| Other | 0 (0) | |
| **Healthcare setting**$^*$ | | |
| Hospital care | 154 (45.3) | |
| Community care | 88 (25.9) | |
| Residential care | 46(13.5) | |
| Mental healthcare | 29 (8.5) | |
| Other / no specific healthcare setting** | 35 (10.3) | |
| **Main field**$^*$ | | |
| Clinical practice | 272 (80.0) | |
| Policy | 67 (19.7) | |
| Education | 41 (12.1) | |
| Research | 28 (8.2) | |
| Politics | 10 (2.9) | |
| | **Mean (SD)** | **Median (min-max)** |
| **Age** (years) | 45.9 (10.6) | 46.0 (24–76) |
| **Expertise in main field** (years) | 19.1 (10.7) | 18 (2–48) |
| **Work related to pharmaceutical care** (hours/week) | 28.3 (15.3) | 30 (1–105) |

$^*$ Total is different from 100% because more than one answer was possible.

** academic setting, education, research, politics, national health services, individual practice (not community care) or not specified

## Responsibility 1: Monitoring therapeutic and adverse effects of medicines

Some respondents considered monitoring patients for the benefits and harms of medicines administered as part of basic nursing care, whereas others disagreed.

**Table 2. Existing or potential nurse responsibilities and tasks in interprofessional pharmaceutical care (beyond medication preparation and administration).**

| Responsibilities | Tasks |
|---|---|
| 1. Monitoring therapeutic and adverse effects of medicines<br>2. Monitoring medicines adherence<br>3. Decision making on medicines use, including (de) prescribing, medication reconciliation and medication review<br>4. Providing patient education and information about medicines | a. Detecting, addressing, reporting clinical change and healthcare problems<br>b. Assessing patients' needs<br>c. Identifying, reporting and addressing drug related problems and safety issues<br>d. Follow-up assessments of patients<br>e. Intervention in emergencies<br>f. Documentation in patient records<br>g. Communication with patient, informal caregiver and family<br>h. Selfcare support and therapeutics education<br>i. Interprofessional communication, including reporting, advising, informing, alerting and discussing<br>j. Communication within the nursing team<br>k. Supervising and coaching new healthcare workers and less qualified team members |

"*I think pharmacists are better placed to report about unwanted effects, since we are committed to report on pharmacovigilance. Pharmaceutical care is pharmacists' work, and nobody else's*"

*(Pharmacist-05, Slovenia)*

"*The clinical evaluation and follow-up is something nurses currently do on a daily basis and which is often the trigger of alarm to physicians. It is already part of nurses' skills and it is being done well.*"

*(Physician-01, Portugal)*

Within monitoring, nurses' tasks were defined as medication anamnesis, detecting clinical change and healthcare problems and assessing patient needs. Early recognition of signals and linking with medicines was seen as vital to patients' safety. Reporting observations to the team (physician and pharmacist) and to patients or their informal caregivers and family, as well as registration and follow-up of medicines' desirable and undesirable effects were recognized as nurses' tasks. Follow-up was suggested as either a nursing or shared responsibility or solely a medical task.

"*Pharmacists won't notice side effects, only one person will–it's the nurse.*"

*(Pharmacist-20, Hungary)*

"*Nurses don't only distribute medicines like a trained monkey. They are able to realize that somehow a problem could arise and preventive interventions might be necessary.*"

*(Nurse-04, Germany)*

To monitor therapeutic and adverse effects of medicines, respondents perceived a certain level of knowledge about medication to be needed, and therefore high quality nurse education must be provided. In addition, some felt clear legal frameworks, policies and regulations, allocating nurses clear roles in monitoring, are necessary.

## Responsibility 2: Monitoring medicines adherence

Many respondents considered adherence monitoring to be a clear and obvious aspect of nurses' roles, while some were convinced that this was a physician-only responsibility or even the sole responsibility of the patient.

> "*Monitoring and following-up medication adherence, this is probably clear. This is an area which is the least controversial, I see no problem in it.*"

> *(Nurse-12, Czech Republic)*

Within monitoring medication adherence, one important nursing task was to detect and alert the interprofessional team of any non-adherence. Nurses may also motivate patients to adhere to their prescribed regimen. Motivational interviewing of patients with targeted open questions would identify reasons of non-adherence, determine patients' needs, and support self-care.

> "*When the nurse is with the patient, she realizes whether the pill is too big for the patient and he would prefer to take two smaller ones twice a day.*"

> *(Pharmacist-02, Italy)*

Prerequisites of adherence monitoring by nurses included: clear guidelines about the responsibilities of nurses, pharmacists and physicians in monitoring adherence within a legal framework; open, blame-free culture; open dialogue between pharmacists, physicians and nurses; appropriate nurse training in PC; and a manageable workload, resulting in time to care and explore issues with patients.

## Responsibility 3: Decision making on medicines use, including (de) prescribing, medication reconciliation and medication review

A wide variation in opinions was reported, with more positive views in countries with existing nurse prescribing. Differences in opinion were not confined to any one profession. A small number of respondents considered nurses already possessed the required competences, and advocated nurse-prescribing for a wide range of medicines, usually within their specialist fields.

> "*Doctors, especially in hospitals, got used to giving their stamps to the head nurse to write prescriptions.*"

> *(Pharmacist-01, Greece)*

> "*We have an internal deal with the nurses on my ward, that they are allowed to give some medicines to patients on their own, under specific circumstances and specific medicines that we agreed on.*"

> *(Physician-06, Slovenia)*

Others favoured nurse prescribing, but only after extra training and under specific conditions, e.g. emergencies, low risk medicines (often those that can be purchased without a prescription) or confined to nurses with high levels of nursing education. A further group would never–under any circumstances—give nurses a role in decision making or prescribing. They

considered this responsibility to be too complex and a medical responsibility, in which collaboration with nurses was not desirable.

> "*Experiences with nurse prescribing in other countries are not of that kind, that we need to be scared of it.*"
>
> *(Pharmacist-02, Belgium)*

> "*It scares me. . . it is probably just my feeling. . . I cannot imagine nurse prescribing.*"
>
> *(Pharmacist-10, Czech Republic)*

Respondents considering decision making on medicines to be a part of nurses' ideal roles described possible tasks within this responsibility: nurses could decide on the route, formulation and brand; add or deprescribe treatments; adjust and titrate doses; prepare prescriptions (to be validated by a physician); and prescribe repeat prescriptions. Respondents predominantly reflected on the selection of products, the level of autonomy and the level of emergency: local and low-risk medicines from a limited list were preferred to systemic and high-risk medicines; supervision by physicians or pharmacists and shared responsibility were favoured above full autonomy for nurses; and life-threatening emergencies warranted increased autonomy. Others felt that more complex thinking is required in these situations, arguing against more responsibility for nurses. There were calls for flexible practice guidelines. Knowledge was mentioned as a crucial prerequisite for decision-making in PC. As an initial step, more pharmacology is needed in pre- and post-registration nurse education. Level 6 (Bachelor) nurses [31], nurse specialists and nurse practitioners were suggested as having the minimum level of education to prescribe.

> "*Nurse prescriptions should be very limited. I would understand nurse prescribing, but only in very specific restricted situations.*"
>
> *(Pharmacist-03, Spain)*

> "*What responsibilities would be part of the ideal role of a nurse in interprofessional pharmaceutical care? In my ambulatory practice I think nurses can prescribe 'repeating prescriptions' within control consultations. I think nurses can decide about routine medicines, within a certain spectrum, within their specialization in the field.*"
>
> *(Physician-22, Slovakia)*

> "*Nurses could have autonomy on the renewal of chronic therapies, previously prescribed by a doctor.*"
>
> *(Nurse-10, Italy)*

> "*Nurses have the right to give emergency therapy when the patient's life is endangered, e.g. in case of major bleeding.*"
>
> *(Nurse-02, Republic of North-Macedonia)*

> "*I would increase the level of knowledge, because if we don't have the proper level of knowledge, we can't prove to doctors and pharmacists that we are competent to prescribe and right now they don't trust us enough to prescribe.*"
>
> *(Nurse-07, Slovenia)*

## Responsibility 4: Providing patient education and information about medicines

Some respondents were convinced that responsibilities for educating and informing patients were the professional territory of physicians or pharmacists, while others believed these responsibilities should be shared with nurses. Opinions were based on the very limited content in pharmacotherapeutics in nurse education.

"*Patient education about medicines would be better done by a pharmacist, they go to school for 5 years and learn everything about medicines, while nurses have only one course in school.*"

*(Pharmacist-01, Slovenia)*

With improved education, nurses could: explain medical diagnoses; inform patients and their caregivers about short- and long-term advantages and disadvantages of their medicines; support self-care; counsel patients at discharge; encourage and empower patients to take their medicines.

"*A nurse has a responsibility to the patient to keep the patient fully informed about what has been prescribed, the risks associated, side effects associated and benefits likewise.*"

*(Physician-04, UK)*

"*I think patient education and providing information is already done, it is common that nurses educate patients. We can discuss about the quality and the way, but I think, the role of nurses should be enhanced here.*"

*(Nurse-12, Czech Republic)*

"*Nurses should provide patient education and information on drugs, because doctors are too complicated for patients.*"

*(Nurse-02, Slovakia)*

## Interactions between nurses, physicians and pharmacists in an ideal interprofessional collaboration

Interprofessional communication, including reporting, advising, informing, alerting and discussing was considered of major importance in interprofessional PC. Collaboration, coaching and supervising within the nursing team was also reported as important.

"*Multidisciplinary communication works, nurses are irreplaceable, they ensure that information and documentation is effectively passed between team members.*"

*(Nurse-03, Slovakia)*

"*Three-dimensional communication is missing. Clinical pharmacists have been collaborating mainly with physicians, discussion with nurses is missing.*"

*(Physician-10, Czech Republic)*

Contextual factors allowed nurses to have a role in interprofessional collaboration, e.g. confidence in nurses' knowledge, self-confidence of nurses, an open blame-free culture, clear roles and responsibilities, availability of team members, involvement of nurses in PC team meetings, absence of hierarchic attitudes, and equality between professionals. Written communication was recommended to ensure proper communication.

"*The working atmosphere is crucial. This must ensure openness and honesty and give room for clear feedback to each other.*"

*(Physician-08, the Netherlands)*

"*I don't know who my nurses are in my two local surgeries. It would be nice to know their names, I don't think that's the nurses' fault I think it's the way we get used to working.*"

*(Pharmacist-24, UK)*

## Strengths, weaknesses, opportunities and threats of nurses' role in interprofessional PC

**Strengths.** The proximity of nurses to patients was a strength of nurses' contribution to PC. Nurses spend a lot of time with patients and these frequent contacts could facilitate screening for symptoms, monitoring adherence, making decisions and informing or educating patients and their informal caregivers.

"*The nurse regularly visits the patient and therefore is the first in line to recognize adverse effects of medicines and to act upon them. Physicians don't spend as much time near the patients' beds, so, they don' t always see the effect of medicines, compared to a nurse on a ward, who walks in the patient's room for about 10 times a day.*"

*(Physician-23, the Netherlands)*

Nurses were seen as well-positioned to take up responsibilities in interprofessional PC. They have key information to share, which can trigger interventions by themselves or other team members, in order to optimize medication use and improve health outcomes. Nurses' reinforcement of physicians' words to patients is important in their role in patient education.

"*I, as a pharmacist, I am a real expert in medication. The GP is an expert in pathology. But nurses, they are 'the eyes' because they SEE patients, they can report to other professionals. Without you, nurses, the healthcare sector is dead. Without you, we are nothing!*"

*(Pharmacist-05, Belgium)*

**Weaknesses.** Firstly, the absence of a legal framework for nurse's roles in PC was evident in several countries. Some professionals reported absence of diagnostic mindsets, PC competences and poor education. Inadequate education promoted a lack of confidence in nurses from some pharmacists, physicians and nurses. Open dialogue with adequate interaction between nurses, pharmacists and physicians seemed to be missing. Although respondents believed that there was more communication than in the past, some hierarchical attitudes persisted.

"*There must be an open dialogue, without throwing remarks, such as 'I am a professional, I am first, you are last.' An open dialogue to be able to say 'Hey guys, who can deal with this part?' It's a puzzle. A brainstorming session to create clear abilities and job descriptions.*"

*(Pharmacist-01, Greece)*

**Opportunities.**   Further, opportunities for nurses' roles in an ideal interprofessional PC were identified. Each professional looking at the patient from his/her own perspective makes the involvement of multiple professionals of added value. Nurse consultations to monitor medicines effects and adherence, and care coordination by nurses were suggested as facilitators of PC. This would align complementary knowledge of team members, and reduce contradictory messages from different professionals.

"*I could not imagine independent prescribing, because of interactions between body systems. A nurse alone cannot order pharmaceuticals, but a team is involved. Each team member has its own perspective; putting knowledge together will lead to much better results.*"

*(Nurse-11, Hungary)*

"*Multidisciplinary teams are the ones who do all the work. It is never a one man's success. Nurses have the capacity to lead, gather and organize multidisciplinary collaborations for the patient's benefit.*"

*(Nurse-02, Greece)*

Nurses taking up more responsibilities in PC could have a positive impact on care quality and patient outcomes: an increase of professional support for patients (including in areas where few physicians are available e.g. rural or post-industrial areas), a substitute for physicians' input, reduction of waiting times and stress for patients, and, in case of nurse prescribing, a facilitation of prescription changes in emergencies.

"*I completely agree that making decisions on medicines would take some weight off doctors shoulders*"

*(Physician-04, Slovenia)*

"*The benefits of interprofessional co-operation with nurses, pharmacists and physicians are rapid response, patient satisfaction and quality of care.*"

*(Nurse-02, Republic of North-Macedonia)*

In addition, shared digital patient files, interprofessional ward rounds and integrating interprofessional collaboration and communication into education of all professionals would be great opportunities for the future.

"*Training with all the professionals is needed, we finish our degree without connecting directly to the other professionals and that is not what we see in the practice.*"

*(Nurse-02, Spain)*

**Threats.**　However, lack of time (to care), shortage of nurses and limited financial compensation for the time spent in PC roles, in combination with the current high burden of nursing responsibilities threaten the realisation of nurses' ideal roles in PC.

> "*I don't understand why things should change, nurses want to prescribe and they don't even have time to do what they are already competent to do…*"
>
> *(Pharmacist-01, Slovenia)*

> "*Those who bear more responsibility should also receive more money, which is not yet the case in today's collective agreements.*"
>
> *(Nurse-06, Germany)*

Finally, the absence of a legal framework for nurses' roles and some physicians or pharmacists worrying about "their territory" in PC must be addressed.

> "*Interactions should be more lubricated and should be encouraged and I think they should be even legislated because it seems that nobody does anything if it is not an obligation… in order to boost public health… but a diagram needs to be made for people understood how it works.. so it will be better to be legislated….*"
>
> *(Pharmacist-06, Portugal)*

> "*The barriers are quite clear, professional conflicts have always been there. Every time one tries to get into a subject to another profession then they put up a stop that "this is my area of responsibility, you shouldn't have anything to do with".*"
>
> *(Nurse-01, Norway)*

> "*My experience is that hospital nurses think they are like physicians and I don't like it. They are also elevated to us as pharmacists, while the role of both our professions is very important. Everybody is better in different area and nobody is the subordinate.*"
>
> *(Pharmacist-02, Slovakia)*

## Discussion

Four main responsibilities for nurses in PC were evaluated. Many different tasks were described as part of nurses' ideal practice, yet many professionals were ambivalent over their implementation.

　　The extent of nursing autonomy depended on type of medicine and country-specific governance structures, and varied from no authority to authority and responsibility for broad ranges of activities. Not every nurse would be capable of performing every task in every situation. Several contextual factors should be taken into account while translating nurses' ideal roles in PC into clinical practice. Important prerequisites which were also already discussed in the literature were: sufficient education [32, 33], knowledge (more pharmacology and pharmacotherapeutics) [34, 35], an interprofessional collaborative approach [1, 36], confidence in nurses [37, 38], an open blame-free culture with clarity of team composition and roles [39, 40], equality between professionals [41], adjusted legislation [42], readiness of professionals and patients to allow nurses to have responsibilities in PC [43], and a manageable workload leaving "time to

care" [44, 45]. Lack of time, shortage of nurses, absent legal frameworks and limited education and knowledge were described as main threats. However, a positive impact on care quality and patient outcomes was associated with nurses taking up responsibilities in PC. Nurses' observations and assessments could convey key patient information to the interprofessional team, as was also shown in previous research [46].

Fourteen countries were included in the study. Despite all of these being in Europe, it cannot be assumed that the education of nurses in each of these countries is uniform. A systematic review of nurse education in European presented differences on both level and duration of education [33]. Two thirds of all nursing education programs are offered at the higher education level, while one third is offered at diploma-level. The duration of full-time nursing education programs varies from two to four years, with the majority (58%) lasting for three years. Also, different education pathways lead to the same level of nursing qualification in some countries and specialist qualifications are offered at both undergraduate and graduate degrees [33]. Although the participants in this study raised the issue of the need for sufficient education before nurses could have a role in pharmaceutical care, experiences on the specific differences between the levels of education in each country were not addressed in the interviews. Only for nurse prescribing did some respondents formulate minimum conditions in terms of educational level. Further research investigating differences in nursing responsibilities between levels of nurse education can offer significant added value to the development of a framework for level-specific roles of nurses in interprofessional PC.Nurses' roles have expanded in Europe over the last decade. An international comparative analysis of reforms of nurse prescribing concluded that 13 European countries already had legislation on nurse prescribing, eight since 2010. The extent of prescribing rights ranged from nearly all medicines within nurses' specialisations to a limited set of medicines. All countries had regulatory and minimum educational requirements in place to ensure patient safety; the majority required some form of physician oversight [47]. Our study included four countries with legal prescribing rights for some nurses or some products at the time of data collection: the Netherlands, Norway, Spain and the United Kingdom. Different participant perspectives, however, were not related to country or any one profession.

Regardless of whether or not nurses are able to prescribe, they can have a pivotal role in initiating and supporting deprescribing [48, 49]. However, nurses' roles in providing patient information about deprescribing are not always well considered, but nurses may be as effective as physicians at discussing medicines discontinuation with patients [50]. When nurses are aware of the medicines that are most appropriate for deprescribing, for example antipsychotics for behaviour disturbance, they can monitor these patients to ascertain the benefits no longer outweigh the harms [48, 49].

We consciously chose to start the interviews with a definition of PC. This strategy has both advantages and disadvantages. Predefining PC ensured uniformity across 14 countries and 12 languages. On the other hand, we were unable to extract the participants' conceptualizations of the definition. However, we did encourage open reflections about the interpretation of role fulfilment within PC. The phenomenological approach of this study incorporates the supposition that there may be multiple truths or realities as perceived by multiple participants [51, 52]. Additionally, the conceptualization of PC responsibilities may differ between healthcare professionals, as was already investigated for the concept of 'medication monitoring' [53]. Monitoring from a nursing perspective is a dynamic, ongoing, day-to-day activity, while pharmacists and physicians typically associate monitoring with structured medication reviews and an intermittent, planned activity [53]. In our study, we were unable to explore any differences in how the concepts or themes were conceptualised by participants. Nevertheless, we described many ambiguous opinions on PC responsibilities and tasks, and participants

elaborated on a broad range of subthemes that needed to be specified in order to define nurses' role in PC.

## Strengths and limitations

To our knowledge, this is the first pan-European qualitative interview study about PC by nurses. The quality of the research can be demonstrated based on the qualitative research quality criteria of Lincloln and Guba [28]. Firstly, triangulation of sources and analyst triangulaton indicate *credibility*. Secondly, the extensive focus on the PC context of the participants resulting in thick descriptions will facilitate *transferability* of the study findings. Thirdly, the *dependability* is confirmed by investigator triangulation: coding of the first interviews by multiple researchers within one country, plus a non-country specific reassessment of the code labels linked to preselected citations by a team of researchers.

The *confirmability* of this research could only be partially achieved. Researchers from all countries were trained in qualitative research, in-depth interviewing, and 'bracketing' their own beliefs about nurses' role in PC during a joint one-week training program. However, since interviewers and respondents often shared work environments, contextual intersecting relationships between the participants and the researchers cannot be ignored. As we wanted to avoid the profession of the researchers influencing the responses from physicians, pharmacists, and other nurses, interviewers were asked not to inform interviewies about their profession unless directly questioned [54].

Another limitation is the absence of structured integration of the field notes, that have been made during the process of transcribing, critical reflecting and coding. Therefore, the researchers might have missed important non-verbal indicators, such as participants' body language and tone of voice.

The selected participants were 'key informant' experts in PC, who knew best what was happening in PC in clinical practice. However, findings cannot be generalised to more junior clinicians or managerial staff. No reimbursement was provided for participation, leading to occasional refusal to participate. Exact numbers of those declining to participate were not registered, leading to an unknown selection bias. Despite the limited number of participants per professional group at national level, no new themes were generated in the last interviews reviewed, suggesting sufficient information power [55]. Socio-cultural influences, mainly in terms of attitudes towards other professions might affect perspectives related to interprofessional collaboration, as was demonstrated in several studies [43, 56]. In this research, no information was sought on cultural and/or ethnic identities of respondents. We wished to avoid sensitive questions and any possibility that respondents might be identified by local readers. Diversity should be taken into account in future research.

## Implications for clinical practice and future research

Our results offer opportunities to create a framework for discussion in clinical practice, collaboration in research, and labour mobility. Nurses, pharmacists and physicians should openly discuss allocation of specific responsibilities and tasks. Our list of responsibilities and tasks is not exhaustive. Medication safety management [57], care coordination [58], overseeing patient medication self-management [59, 60], assessing patients' competences [61], coaching and training patients [62], discharge planning [63] and interprofessional referrals [64] are additional nursing responsibilities and tasks identified in the literature. A scoping review of research about PC by nurses would be useful to confirm the completeness of the role described or supplement with additional responsibilities and tasks. Further research should also address the differences in nurses' roles within different levels of nurse education.

Exploring nurses' ideal role in PC is not intended to remove responsibilities from other professional groups. On the contrary, the benefits of interprofessional collaboration and communication between pharmacists, physicians and nurses and its major impact on care quality and patient outcomes have already been amply demonstrated [48, 49, 65–69]. Yet, healthcare systems are historically hierarchical in nature with physicians regularly assuming leadership positions and decision-making roles. Frustrations, lack of confidence, lack of organization and structural hierarchies hinder interprofessional relationships and communication [41]. Power imbalance between professions is an important factor in nurses' professional roles when discussing PC and its formalisation. To address this source of conflict, it may be helpful for team members to discuss and agree roles and responsibilities [40]. Increasing the awareness of all team members' potential roles would allow pharmacists, nurses and physicians to benefit from teamwork [65]. Also, educators hesitate to address the reality of hierarchies in healthcare [70]. The training of healthcare professionals remains largely single discipline, which may reduce the ability to collaborate interprofessionally [71]. Therefore, we call for more interprofessional education, as well as rigorous research on interprofessional PC to tackle the remaining barriers.

## Conclusion

Nurses have an active role in monitoring patients for the impact of their medicines, monitoring adherence, making decisions on medicines, and providing patient education and information. Different tasks within these responsibilities have been described, although contextual, knowledge and training factors have to be considered before nurses can perform this ideal role. Lack of time, shortage of nurses, an absent legal framework and limited education and knowledge were the main threats for nurses' roles in PC. Nevertheless, a positive impact on care quality and patient outcomes was associated with nurses taking up responsibilities in PC. Nurses' observations and assessments could lead to key information about patients being shared and addressed by the interprofessional team. The outcomes of this study evidence the need for a consensus-based PC framework across Europe.

## Supporting information

**S1 Appendix. Interview guide of the interview study in 14 countries.** English version and 12 translations.
(PDF)

**S2 Appendix. Acknowledgements: List of interviewers, manuscript reviewers, people facilitating access to the field and other contributors.**
(PDF)

**S3 Appendix. Demographics database.**
(XLSX)

**S1 Table. Code book of the interview study.** Codes S001 to S042 were the codes of the first code book, codes S043 to S049 were added to the final code book.
(PDF)

## Acknowledgments

The authors explicitly thank all interviewers for their valuable contribution in collecting data and other contributors for reviewing the manuscript and giving us access to the field (S3 Appendix).

## Author Contributions

**Conceptualization:** Elyne De Baetselier, Tinne Dilles, Nienke E. Dijkstra, Carolien G. Sino, Bart Van Rompaey.

**Data curation:** Elyne De Baetselier, Tinne Dilles, Bart Van Rompaey.

**Formal analysis:** Elyne De Baetselier, Tinne Dilles, Luis M. Batalha, Nienke E. Dijkstra, Maria I. Fernandes, Izabela Filov, Juliane Friedrichs, Vigdis A. Grondahl, Jana Heczkova, Ann Karin Helgesen, Sue Jordan, Sarah Keeley, Thomas Klatt, Petros Kolovos, Veronika Kulirova, Sabina Ličen, Manuel Lillo-Crespo, Alba Malara, Hana Padysakova, Mirko Prosen, Dorina Pusztai, Jorge Riquelme-Galindo, Jana Rottkova, Carolien G. Sino, Francesco Talarico, Styliani Tziaferi, Bart Van Rompaey.

**Funding acquisition:** Elyne De Baetselier, Tinne Dilles, Bart Van Rompaey.

**Investigation:** Elyne De Baetselier, Tinne Dilles, Luis M. Batalha, Nienke E. Dijkstra, Maria I. Fernandes, Izabela Filov, Juliane Friedrichs, Vigdis A. Grondahl, Ann Karin Helgesen, Thomas Klatt, Petros Kolovos, Veronika Kulirova, Sabina Ličen, Manuel Lillo-Crespo, Alba Malara, Hana Padysakova, Mirko Prosen, Dorina Pusztai, Jorge Riquelme-Galindo, Jana Rottkova, Carolien G. Sino, Francesco Talarico, Styliani Tziaferi, Bart Van Rompaey.

**Methodology:** Elyne De Baetselier, Tinne Dilles, Nienke E. Dijkstra, Jana Heczkova, Sue Jordan, Sarah Keeley, Carolien G. Sino, Bart Van Rompaey.

**Project administration:** Elyne De Baetselier, Tinne Dilles, Bart Van Rompaey.

**Resources:** Elyne De Baetselier, Tinne Dilles, Bart Van Rompaey.

**Software:** Elyne De Baetselier, Tinne Dilles, Bart Van Rompaey.

**Supervision:** Elyne De Baetselier, Tinne Dilles, Bart Van Rompaey.

**Validation:** Elyne De Baetselier, Tinne Dilles, Bart Van Rompaey.

**Visualization:** Elyne De Baetselier, Tinne Dilles, Bart Van Rompaey.

**Writing – original draft:** Elyne De Baetselier.

**Writing – review & editing:** Elyne De Baetselier, Tinne Dilles, Luis M. Batalha, Nienke E. Dijkstra, Maria I. Fernandes, Izabela Filov, Juliane Friedrichs, Vigdis A. Grondahl, Jana Heczkova, Ann Karin Helgesen, Sue Jordan, Sarah Keeley, Thomas Klatt, Petros Kolovos, Veronika Kulirova, Sabina Ličen, Manuel Lillo-Crespo, Alba Malara, Hana Padysakova, Mirko Prosen, Dorina Pusztai, Jorge Riquelme-Galindo, Jana Rottkova, Carolien G. Sino, Francesco Talarico, Styliani Tziaferi, Bart Van Rompaey.

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
