## [Decision Letter · Decision Letter 0]

24 Feb 2021

PONE-D-20-36633

Perspectives about nurses' role in interprofessional pharmaceutical care across 14 European countries: a qualitative study in pharmacists, physicians and nurses

PLOS ONE

Dear Dr. De Baetselier,

Thank you for submitting your manuscript to PLOS ONE. After careful consideration, we feel that it has merit but does not fully meet PLOS ONE’s publication criteria as it currently stands. Therefore, we invite you to submit a revised version of the manuscript that addresses the points raised during the review process.

In addition to the reviewer comments, please consider the following:

General comments

- Please ensure the manuscript conforms to qualitative research reporting standards e.g. COREQ guidelines.

- Please comment on how qualitative research quality criteria of credibility, applicability/transferability, consistency/dependability, neutrality/confirmability (adapted from Lincoln and Buga, 1985) are considered in your methods e.g. triangulation, member checking, 'thick' description and/or other techniques.

- Please pay careful attention to Review 1's comment of the profession of the researchers and how that may influence how the research was conducted (reflexivity)

- A phenomenological approach was undertaken which incorporates the supposition that there may be multiple truths as perceived by multiple participants. Throughout the results, were there any differences in how the concepts/themes were conceptualised by the participants? For instance, we identified that nurses conceptualised 'monitoring' of medication differently to pharmacists and physicians (Langford AV, Ngo GT, Chen TF, Roberts C, Schneider CR. Nurses', Pharmacists' and Family Physicians' Perceptions of Psychotropic Medication Monitoring in Australian Long-Term Care Facilities: A Qualitative Framework Analysis. Drugs & Aging. 2020 Dec 14.) Differences or commonality in conceptualisation of concepts such as pharmaceutical care, monitoring, adherence as explored by this research, across professions and practice settings/countries, would be of interest to the audience. This also may speak to Reviewer 1's comment on expanding beyond the initial (gold standard) definition of pharmaceutical care by Hepler&Strand. Do participants' conceptualisations align with this definition or diverge? In what ways?

Specific comments:

- consider changing the title to "Perspectives of nurses' roles in interprofessional pharmaceutical care across 14 European countries: a qualitative study in pharmacists, physicians and nurses"

- line 184: consideration of reliability is made. Reliability is primarily a positivist concept, however a phenomenological paradigm was adopted. There appears to be an inherent paradigmatic contradiction apparent. Please review.

- line 513-515: please provide supporting evidence for these statements

Overall, this is an important piece of work, and thank you for the submission to PLOS ONE, it is a pleasure to be the Academic Editor of this manuscript.

We look forward to receiving your revised manuscript.

Kind regards,

Carl Richard Schneider, BN, BPharm (Hon), PhD

Academic Editor

PLOS ONE

Journal Requirements:

2. During our internal checks, the in-house editorial staff noted that you conducted research or obtained samples in another country. Please check the relevant national regulations and laws applying to foreign researchers and state whether you obtained the required permits and approvals. Please address this in your ethics statement in both the manuscript and submission information. In addition, please ensure that you have suitably acknowledged the contributions of any local collaborators involved in this work in your authorship list and/or Acknowledgements. Authorship criteria is based on the International Committee of Medical Journal Editors (ICMJE) Uniform Requirements for Manuscripts Submitted to Biomedical Journals - for further information please see here: https://journals.plos.org/plosone/s/authorship.

3. Please include a copy of the interview guide used in the study, in both the original language and English, as Supporting Information, or include a citation if it has been published previously.

4. Thank you for stating the following in the Financial Disclosure section:

"The research was supported by the Erasmus+ Programme of the European Union [grant number 2018-1-BE02-KA203-046861] and MDMJ accountants, an accountancy service in Belgium that financially supported the Belgian authors (www.mdmj.be). The funders had no role in study design, data collection and analysis, decision to publish, or preparation of the manuscript."

We note that you received funding from a commercial source: MDMJ accountants.

6. We note that Figure 1 in your submission contain map images which may be copyrighted. All PLOS content is published under the Creative Commons Attribution License (CC BY 4.0), which means that the manuscript, images, and Supporting Information files will be freely available online, and any third party is permitted to access, download, copy, distribute, and use these materials in any way, even commercially, with proper attribution. For these reasons, we cannot publish previously copyrighted maps or satellite images created using proprietary data, such as Google software (Google Maps, Street View, and Earth). For more information, see our copyright guidelines: http://journals.plos.org/plosone/s/licenses-and-copyright.

6.1.    You may seek permission from the original copyright holder of Figure 1 to publish the content specifically under the CC BY 4.0 license. 

6.2.    If you are unable to obtain permission from the original copyright holder to publish these figures under the CC BY 4.0 license or if the copyright holder’s requirements are incompatible with the CC BY 4.0 license, please either i) remove the figure or ii) supply a replacement figure that complies with the CC BY 4.0 license. Please check copyright information on all replacement figures and update the figure caption with source information. If applicable, please specify in the figure caption text when a figure is similar but not identical to the original image and is therefore for illustrative purposes only.

Reviewers' comments:

Reviewer's Responses to Questions

**Comments to the Author**

1. Is the manuscript technically sound, and do the data support the conclusions?

Reviewer #1: Partly

Reviewer #2: Yes

2. Has the statistical analysis been performed appropriately and rigorously? 

Reviewer #1: N/A

Reviewer #2: Yes

3. Have the authors made all data underlying the findings in their manuscript fully available?

Reviewer #1: Yes

Reviewer #2: Yes

4. Is the manuscript presented in an intelligible fashion and written in standard English?

Reviewer #1: Yes

Reviewer #2: Yes

5. Review Comments to the Author

Reviewer #1: Dear Authors

Thank you for submitting this interesting manuscript. It is generally well written, however, in my opinion, there are several points for development. These are listed below:

1. The study centres on pharmaceutical care and the perceived actual or potential role that nurses play in this. Hepler and Strand's 1990 (page 4) definition is used to provide the context to this, yet, this definition could be considered both dated and limited in its scope. Is there a more recent definition that could be used?

2. I am uncertain what the purpose of establishing a framework for nurses' roles in interprofessional pharmaceutical care is, and this is not explained in the manuscript. Perhaps, some background discussion of the benefits of this or why it is needed would be helpful.

3. It is stated that the study is part of the DeMoPhaC project, yet very little information is provided about this. For a reader who has not read previous reports of this project, some additional information would be useful.

4. It is clearly identified that a phenomenological case study design has been used as the study approach. Again, more detail of how this was applied as well as justification for its use is needed.

5. Fourteen (14) countries were included in the study. Despite all of these being in Europe, it cannot be assumed that the education and roles of the registered nurses in each of these countries is uniform. This needs to be discussed in light of both the findings and the discussion. Not doing this is a considerable weakness of the study.

6. Explanation of the recruitment process lacks detail. How were participants recruited and informed about the study- who provided this information and how?

7. How was the sample size determined? It is stated that "We aimed for at least two interviews per profession (no.3) per healthcare setting". How was this number decided on? Was the issue of data saturation considered?

8. Throughout the manuscript the phrase "nurses potential and/or ideal ideal roles in pharmaceutical care" is used. It is unclear what is meant by these descriptors, which are very subjective and somewhat meaningless. Are the authors' referring to nurses "scope of practice" in pharmaceutical care ?

9. I found a lack of clarity/ distinction between what is considered a role, responsibility or task, and there is clearly some overlap. For example" Providing patient education and information about medications could be considered all three. Consideration around choice of words is recommended.

10. The Discussion section is brief and limited. Some important points have been alluded to, however these have not been followed up/ explored with adequate discussion or links to the literature. For example: the very long sentence on page 23 (lines 494-499) states: "Most important pre-requisites were: sufficient education, knowledge (more pharmacology and pharmacotherapeutics), an interprofessional collaborative approach, confidence in nurse, an open blame- free culture with clarity of team composition and roles, equality between professionals, adjusted legislation, readiness of professionals and patients to allow nurses to have responsibilities in PC, and a manageable workload leaving "time to care". There are some very important issues that have been identified but not discussed or elucidated.

11. There are several minor grammatical and punctuation issues throughout that careful proof-reading may assist to identify.

12. Table 1 has some additional, misplaced numbers (211, 212, 213, 214) that need removal.

Reviewer #2: The purpose of this study was to assess the role of the nurse in pharmaceutical care in European healthcare settings based on the perspective of pharmacists, physicians, and other nurse healthcare professionals. Overall, this paper was informative and comprehensive with good emphasis on balanced professional diversity. I particularly appreciated how the different viewpoints of pharmacists, physicians, and nurses were captured in the paper. Specifically, the role of nurses in pharmaceutical care can be a controversial topic as the authors describe, and I appreciated that the paper captured many opinions and weaknesses and strengths on the topic.

Suggested changes are described below:

1. (lines 145, 454): The term interprofessional should be used rather than multidisciplinary. Multidisciplinary refers to activities performed by members of different academic disciplines. As previously defined, the term interprofessional should be used in this healthcare context. (see https://interprofessional.global/wp-content/uploads/2019/10/Guidance-on-Global-Interprofessional-Education-and-Collaborative-Practice-Research_Discussion-Paper_FINAL-WEB.pdf)

2. (line 205) The text of the results section, reads that 43% of the participants were employed in hospital care and 24% of the participants were employed in community care, however, this is not consistent with the results in Table 1 in the Healthcare settings section (48.6% and 27.0%, respectively).

Also, regarding that same section (Healthcare settings) in Table 1, the n-values total 313 rather than at least 340 interviews. What is the reason for this number being less than the total interviews? Did some people not respond to the question? Why would it not be 340 or greater particularly given that many of these healthcare professionals may be employed in more than one healthcare setting?

3. (line 165-167) As interviews were conducted in the workplace or an alternative location, can you comment on participant body language, tone of voice, and/or other non-verbal cues that might be important indicators to the responses to the research questions?

4. Did you collect information about cultural and/or ethnic diversity of the study participants? Did you consider that some participants might identify with a gender other than male or female? Could either or both cultural/ethnic background or gender identity affect perspectives related to interprofessional collaboration in pharmaceutical care?

5. Limitations:

i) (line 518) As the interviews were conducted by nurses, this could be a limitation as this may have potentially affected the responses from physicians, pharmacists, and other nurses. This limitation should be mentioned in the limitations section of the paper.

6. Overall, there are a few formatting errors such as in Table 1 the line numbers 211-214 are inside the cell.

Also, the writing could be further edited for clarity and conciseness for the reader.

6. PLOS authors have the option to publish the peer review history of their article (what does this mean?). If published, this will include your full peer review and any attached files.

Reviewer #1: No

Reviewer #2: **Yes: **Kathleen M. MacMillan

---

## [Author Response · Author response to Decision Letter 0]

26 Mar 2021

26 March 2021

Dear editor, dear reviewers,

Thank you for reviewing our manuscript (PONE-D-20-36633) entitled “Perspectives of nurses' role in interprofessional pharmaceutical care across 14 European countries: a qualitative study in pharmacists, physicians and nurses". Your comments have allowed us to improve the content and the clarity of the manuscript. The changes we made to our original manuscript are documented below.

General comments

- Please ensure the manuscript conforms to qualitative research reporting standards e.g. COREQ guidelines.

In our first submission we didn’t mention this reporting standard, although it was used to report the study. We have added the following sentence to the methods: “This study was conducted and reported according to the Consolidated Criteria for Reporting Qualitative Research (COREQ).” We also added the appropriate reference. 

- Please comment on how qualitative research quality criteria of credibility, applicability/transferability, consistency/dependability, neutrality/confirmability (adapted from Lincoln and Guba, 1985) are considered in your methods e.g. triangulation, member checking, 'thick' description and/or other techniques.

Thank you for drawing our attention to the lack of a clear description of these quality criteria. In our revised manuscript, we have elaborated on these concepts in the discussion (strengths and limitations) and referred to Lincoln and Guba (1985).

* Credibility

 Triangulation of sources: In this study, people with different perspectives have been interviewed: three different professional groups in four healthcare settings

 Analyst triangulation: In each country, every first interview for each professional group (nurses, physicians, pharmacists) was analysed by two researchers independently. In that way, at least three interviews for each of the 14 countries were analysed by two researchers independently. 

 Coding of all other transcripts was checked by co-authors in each country. After all interviews were coded, two researchers of the Belgian team (EDB, TD) reviewed all preselected and translated citations to assess the code labels for accuracy.

* Transferability

 The interviews consisted of open-ended questions, focusing on the pharmaceutical care context of the healthcare workers. The resulting thick descriptions will facilitate transferability of findings.

* Dependability 

 Investigator triangulation was achieved by multiple researchers within each country coding the first interviews, plus a non-country specific reassessment of the code labels linked to preselected citations by a team of researchers.

* Confirmability

 Triangulation of sources and analyst triangulation as described above.

 Reflexivity: Researchers in all countries were trained in qualitative research, in-depth interviewing and ‘bracketing’ their own beliefs about nurses’ role in pharmaceutical care during a joint one-week training program. However, since interviewers and respondents often shared work 

 environments, potential contextual intersecting relationships between the participants and the researchers cannot be ignored. As we wanted to avoid the profession of the researchers influencing the responses from physicians, pharmacists, and other nurses, interviewers were asked not to 

 inform interviewees about their profession unless directly questioned. By acknowledging this limitation in the discussion, we aimed to establish the transparency in our study.

- Please pay careful attention to Review 2's comment of the profession of the researchers and how that may influence how the research was conducted (reflexivity)

Thank you for this comment. As indicated in the answer on the previous comment, we have addressed this limitation transparently in the discussion.

- A phenomenological approach was undertaken which incorporates the supposition that there may be multiple truths as perceived by multiple participants. Throughout the results, were there any differences in how the concepts/themes were conceptualised by the participants? For instance, we identified that nurses conceptualised 'monitoring' of medication differently to pharmacists and physicians (Langford AV, Ngo GT, Chen TF, Roberts C, Schneider CR. Nurses', Pharmacists' and Family Physicians' Perceptions of Psychotropic Medication Monitoring in Australian Long-Term Care Facilities: A Qualitative Framework Analysis. Drugs & Aging. 2020 Dec 14.) Differences or commonality in conceptualisation of concepts such as pharmaceutical care, monitoring, adherence as explored by this research, across professions and practice settings/countries, would be of interest to the audience. This also may speak to Reviewer 1's comment on expanding beyond the initial (gold standard) definition of pharmaceutical care by Hepler&Strand. Do participants' conceptualisations align with this definition or diverge? In what ways?

Thank you for these considerations. We have refined our methods and discussion based on this comment. 

We consciously chose to start the interviews with a definition of PC. This strategy has both advantages and disadvantages. Predefining PC ensured uniformity across 14 countries and 12 languages. On the other hand, we were unable to extract the participants’ conceptualizations of the definition. However, we did encourage open reflections about the interpretation of role fulfilment within PC. We have added to the methodology the PC definition that was presented to the participants: “To ensure conformity across twelve languages, the concept of PC was described at the beginning of the interview: “healthcare professionals’ contribution to the care of individuals in order to optimize medicines use and improve health outcomes”. This description was derived from the Pharmaceutical Care Network Europe definition of 2013, taking into account the interprofessional aspect of PC, as recently (2020) acknowledged by the New Council of Europe resolution to promote pharmaceutical care in Europe .[references are added in the reference list]”

In the discussion we have elaborated on our choice to start the interviews with the same definition of PC for all participants. We acknowledged the ‘multiple truths’ within our phenomenological approach and referred to the example of ‘medication monitoring’, which is differently conceptualized by nurses, physicians and pharmacists. Thank you for this reference.

In our study, we were unable to explore any differences in how the concepts or themes were conceptualised by participants. Nevertheless, we described many ambiguous opinions on PC responsibilities and tasks, and participants elaborated on a broad range of subthemes that needed to be specified in order to define nurses’ role in PC.

Specific comments

- Consider changing the title to "Perspectives of nurses' roles in interprofessional pharmaceutical care across 14 European countries: a qualitative study in pharmacists, physicians and nurses"

Thank you for this comment. We changed “Perspectives about nurses’ role” into “Perspectives of nurses’ role”.

- Line 184: consideration of reliability is made. Reliability is primarily a positivist concept, however a phenomenological paradigm was adopted. There appears to be an inherent paradigmatic contradiction apparent. Please review.

We acknowledge that reliability, as a positivistic concept, was not the best choice of terminology. We have replaced “reliability” by “confirmability” and we also adjusted the citation. We cited Lincoln & Guba (1985), Tracy (2010) and Korstjens & Moser (2018) to support the use of ‘confirmability’.

- Line 513-515: please provide supporting evidence for these statements

We have now cited the study of Wright, Scott, Buck, et al (2019).

Additional Journal Requirements 

We have carefully considered the formatting of the documents and the file naming to ensure that all PLOS ONE’s style requirements are met.

2. During our internal checks, the in-house editorial staff noted that you conducted research or obtained samples in another country. Please check the relevant national regulations and laws applying to foreign researchers and state whether you obtained the required permits and approvals. Please address this in your ethics statement in both the manuscript and submission information. In addition, please ensure that you have suitably acknowledged the contributions of any local collaborators involved in this work in your authorship list and/or Acknowledgements. Authorship criteria is based on the International Committee of Medical Journal Editors (ICMJE) Uniform Requirements for Manuscripts Submitted to Biomedical Journals - for further information please see here: https://journals.plos.org/plosone/s/authorship.

First, the study protocol was approved by the Ethics Committee for Social Sciences and Humanities of the University of Antwerp. Then, the researchers in each of the participating countries checked the research ethics requirements in their own countries. Based on local regulations, additional approval was needed in the UK, Slovenia and Portugal. None of the other 10 countries requested additional ethical approval for this interview study.

The three Research Ethics Committees are mentioned in the last section of the methods. We clarified that “national regulations and laws of the other countries didn’t require other permits or approvals.”

No-one collected data outside their own country.

3. Please include a copy of the interview guide used in the study, in both the original language and English, as Supporting Information, or include a citation if it has been published previously.

The English version of the interview guide is in the Supporting information ‘S1 Appendix’. This has now been augmented by the translations into all 12 other original languages: Dutch (Belgium and the Netherlands), Czech, German, Greek, Hungarian, Italian, Macedonian, Norwegian, Portuguese, Slovak, Slovenian and Spanish.

4. Thank you for stating the following in the Financial Disclosure section:

"The research was supported by the Erasmus+ Programme of the European Union [grant number 2018-1-BE02-KA203-046861] and MDMJ accountants, an accountancy service in Belgium that financially supported the Belgian authors (www.mdmj.be). The funders had no role in study design, data collection and analysis, decision to publish, or preparation of the manuscript."

We note that you received funding from a commercial source: MDMJ accountants.

We have added an amended Competing Interest Statement, explicitly mentioning MDMJ accountants as a commercial funder of this study, and we have included the statement in our cover letter.

Ethical restrictions have been imposed on data sharing by the Ethics Committee for Social Sciences and Humanities of the University of Antwerp, the Medical Ethics Committee of the Republic of Slovenia and the UK NHS Research Ethics Committee that approved this study. The data contain potentially identifying and sensitive information. Also, investigators from the other countries confirmed that making these sensitive data publicly available without having requested the consent of the interviewees beforehand, is impossible for ethical and legal concerns. 

1) UK data are stored at Swansea University, Swansea, UK. All proposals to view the data are subject to review by Swansea University’s Research Governance department and the PI. Before any data can be accessed, approval must be given. 

The application process is via the Academic Lead for Research Integrity Research Engagement & Innovation Services, Swansea University and the PI or Neil Carter.

Contacts: Swansea University, Swansea SA2 8PP

Tel: +44 /0 1792 606060 and 518541 or 295610

Email: researchgovernance@swansea.ac.uk, s.e.jordan@swansea.ac.uk or n.carter@swansea.ac.uk

2) Data from the other 13 countries are stored at the University of Antwerp, Antwerp, Belgium.

All proposals to view the data are subject to review by University of Antwerp’s Research Governance department and the PI. Before any data can be accessed, approval must be given. 

The application process is via the data protection officer, Antwerp University and the PI

Contacts: University of Antwerp, Prinsstraat 13, 2000 Antwerp, Belgium.

Email: privacy@uantwerpen.be, Tinne.Dilles@uantwerpen.be

There are no restrictions on sharing the de-identified demographic data, which we have uploaded as a supplementary file.

6. We note that Figure 1 in your submission contain map images which may be copyrighted. 

Thank you for you comment. After careful consideration, we decided to remove the copyrighted map from Figure 1. The adjusted figure is added to this submission.

5. Review Comments to the Author

Reviewer #1:

1. The study centres on pharmaceutical care and the perceived actual or potential role that nurses play in this. Hepler and Strand's 1990 (page 4) definition is used to provide the context to this, yet, this definition could be considered both dated and limited in its scope. Is there a more recent definition that could be used?

Thank you for this comment. In 2013, the Pharmaceutical Care Network Europe (PCNE) held an expert consensus meeting, during which a new European definition for Pharmaceutical Care was created, based on Hepler and Strand’s definition from 1990. Both definitions, however, were limited to the contribution of pharmacists. More recently, in March 2020, the New Council of Europe resolution to promote pharmaceutical care in Europe, broadened the definition and acknowledged the need for interprofessional collaboration in pharmaceutical care. In our revised manuscript, we have referred to this more recent definition, which is less limited in its scope. Based on the original definition of Hepler and Strand, the adjusted definition of PCNE and the extended definition of the New Council of Europe resolution, we have used the following definition of pharmaceutical care in the interviews, as well as in the manuscript: ‘healthcare professionals’ contribution to the care of individuals in order to optimize medicines use and improve health outcomes. (Hepler and Strand, 1990; Allemann, 2013; Council of Europe, 2020)

2. I am uncertain what the purpose of establishing a framework for nurses' roles in interprofessional pharmaceutical care is, and this is not explained in the manuscript. Perhaps, some background discussion of the benefits of this or why it is needed would be helpful.

Thank you for this comment, which indicates that we didn’t sufficiently elaborate on the importance of the proposed framework. Because of the existing obfuscation of role boundaries, collaboration on different levels is hindered: quality of interprofessional communication and collaboration in daily clinical practice; transnational collaboration in research, education and innovation; and labour mobility of nurses. We want to investigate which responsibilities nurses are allowed to assume, and which responsibilities nurses are able to assume (legally or otherwise). In the introduction, we mentioned that ‘a framework for nurses’ ideal roles in interprofessional pharmaceutical care would facilitate discussions in clinical practice, education, research, international comparisons, policy-making and legislation.’ We added that 1) a framework would allow insights into current and potential roles and 2) this framework could be used to: develop an assessment to evaluate nurses’ competences in pharmaceutical care, guide evaluation of nurse education, support nurse educators, benchmark practice standards, and underpin nurses’ labour mobility.

3. It is stated that the study is part of the DeMoPhaC project, yet very little information is provided about this. For a reader who has not read previous reports of this project, some additional information would be useful.

During this revision, we have elaborated on the overall DeMoPhaC project and its overall aim in the introduction: ‘The DeMoPhaC project is an international Erasmus+ collaboration to investigate nurses’ role in interprofessional pharmaceutical care in 14 countries. Within this project, several large-scale quantitative and qualitative studies are being undertaken with healthcare workers and nursing students. The overall aim of the project is the development of nursing curricula and final year nursing students’ competences in pharmaceutical care.’

4. It is clearly identified that a phenomenological case study design has been used as the study approach. Again, more detail of how this was applied as well as justification for its use is needed.

Thank you. In the methods section we expanded on how and why a phenomenological approach was adopted, together with the appropriate references: “Phenomenology is well suited for exploring perspectives of healthcare professionals. This research approach was chosen as an appropriate way to describe the essence of the phenomenon “nurses’ role in interprofessional PC”, by exploring it from the perspective of those who have experienced it, namely pharmacists, physicians and nurses themselves. Interviewing this study population facilitates studying and understanding healthcare professionals’ lived experiences in interprofessional PC. Only by understanding their personal experiences and perceptions of nurses’ responsibilities and tasks, and interprofessional collaboration and communication, we will be able to provide detailed examination of the current strengths and weaknesses, together with the future opportunities and threats from nurses’ involvement in PC.” (Neubauer, 2019)

5. Fourteen (14) countries were included in the study. Despite all of these being in Europe, it cannot be assumed that the education and roles of the registered nurses in each of these countries is uniform. This needs to be discussed in light of both the findings and the discussion. Not doing this is a considerable weakness of the study.

We acknowledge the importance of discussing the differences in education and roles of nurses throughout Europe. Before starting this study, we mapped the levels of nursing education in the participating countries. Despite theoretically comparable levels, the quality and the PC-related content could not be evaluated. In this study, the focus was on healthcare workers’ expectations of nurses in clinical practice. This should allow the needs of practice to be more closely aligned with nurse education. Future research within the DeMoPhaC project will address the differences in nurses’ roles within different levels of nurse education. Also, we elaborated on the importance of discussing the differences in education and roles of nurses throughout Europe in the discussion: “A systematic review of nurse education in Europe presented differences on both level and duration of education. Two thirds of all nursing education programs are offered at the higher education level, while one third is offered at diploma-level. The duration of full-time nursing education programs varies from two to four years, with the majority (58%) lasting for three years. Also, different education pathways lead to the same level of nursing qualification in some countries, and specialist qualifications are offered as both undergraduate and graduate degrees. Although the participants in this study raised the issue of the need for sufficient education before nurses could have a role in pharmaceutical care, experiences on the specific differences between the levels of education in each country were not addressed in the interviews. Only for nurse prescribing did some respondents formulate minimum conditions in terms of educational level. Further research investigating differences in nursing responsibilities between levels of nurse education can offer significant added value to the development of a framework for level-specific roles of nurses in interprofessional PC.”

6. Explanation of the recruitment process lacks detail. How were participants recruited and informed about the study, who provided this information and how?

We have added the following explanation in the methods: “Representatives of professional associations for nurses, physicians and pharmacists and health care providers in different healthcare institutions were asked to identify key informants. Researchers contacted the persons, identified as potential participants, by email or telephone, informed them about the study, and about being named as a key informant on nurses’ role in interprofessional PC. If they agreed with being able to serve as a key informant, written information was provided to fully inform the potential participants about the study details.”

7. How was the sample size determined? It is stated that "We aimed for at least two interviews per profession (no.3) per healthcare setting". How was this number decided on? Was the issue of data saturation considered?

This sample size was chosen in order to compile a sample with perspectives as diverse as possible. Therefore, as a part of ‘triangulation of sources’, we aimed for more than one interview per professional group in each of four main healthcare settings (hospital care, ambulatory community care, residential care and mental healthcare). We estimated that this sample size would lead to data saturation. We have added the sample size determination in the methods.

Yes, the issue of data saturation was considered and reached in all countries. We have now mentioned this in the manuscript (methods).

8. Throughout the manuscript the phrase "nurses potential and/or ideal roles in pharmaceutical care" is used. It is unclear what is meant by these descriptors, which are very subjective and somewhat meaningless. Are the authors' referring to nurses "scope of practice" in pharmaceutical care?

Thank you. In the revised manuscript we have clarified the meaning of “nurses’ potential or ideal roles”. We consider nurses’ scope of practice as the full range of roles, responsibilities and tasks that nurses are educated, competent and authorized to perform. Within this scope of practice we also want to look beyond the legal framework. This is a combination of nurses’ current scope of practice and the future/potential/ideal scope of practice. In clinical situations we cannot assume that current roles correspond with how all healthcare workers’ would like this role to be fulfilled. Our focus is on quality of care and how nurses could achieve the best patient outcomes. In a previous quantitative study, we described nurses’ current practices in pharmaceutical care; in this study we wanted to extend the current practice with the perspectives of healthcare workers about the ideal role of a nurse. (For example, nurse prescribing is legal in some countries. In other countries nurses do prescribe, without this being a part of their legal scope of practice. Therefore, nurse prescribing is part of the potential/ideal role of nurses in countries without a current legal framework or without other prerequisites to allow nurses to prescribe. (De Baetselier, et al. 2020))

We have added 1) a description of ‘scope of practice’: Nurses’ scope of practice is considered as the full range of roles, responsibilities and tasks that nurses are educated, competent and authorized to perform (CMA, 2003; College of Licensed Practical Nurses of Alberta, 2003) and 2) the following sentence in the introduction (aim) of the revised manuscript: “By considering the ‘potential or ideal roles’, we aimed to investigate nurses’ responsibilities and tasks within – but also beyond – nurses’ current legal scope of practice, taking into account all necessary contextual factors.

9. I found a lack of clarity/ distinction between what is considered a role, responsibility or task, and there is clearly some overlap. For example" Providing patient education and information about medications could be considered all three. Consideration around choice of words is recommended.

Thank you for pointing out this obfuscation. In the methods, we have added the following paragraph together with the appropriate references to the literature: “Responsibilities and tasks were defined based on the literature, together with discussions with an expert in health law, liability law and ethics and an expert in legal philosophy and ethics: ‘The role of nurses involves several responsibilities. A responsibility for nurses is an obligation that they have by virtue of their role as a nurse. Their central responsibility is to be the patient’s health advocate and to provide high quality care, using sound professional judgement and taking into account the relevant legal and moral considerations. The other responsibilities of nurses derive from this central responsibility. Nurses can be made to answer for failing in their responsibilities, which could result in disciplinary, civil, and criminal liability. Specific tasks may have to be performed in order to fulfill a responsibility.’” (Nursing and Midwifery Board of Ireland, 2015; Krautscheid, 2004)

10. The Discussion section is brief and limited. Some important points have been alluded to, however these have not been followed up/ explored with adequate discussion or links to the literature. For example: the very long sentence on page 23 (lines 494-499) states: "Most important pre-requisites were: sufficient education, knowledge (more pharmacology and pharmacotherapeutics), an interprofessional collaborative approach, confidence in nurse, an open blame- free culture with clarity of team composition and roles, equality between professionals, adjusted legislation, readiness of professionals and patients to allow nurses to have responsibilities in PC, and a manageable workload leaving "time to care". There are some very important issues that have been identified but not discussed or elucidated.

Thank you. The various discussion elements, raised by the editor and both reviewers, were explored in greater depth, which allowed us to improve the content and the clarity of the discussion. 

The following changes were made: we linked literature to the different contextual factors that were extracted from the interviews; we elaborated on the differences in education and roles of nurses throughout Europe; we acknowledged the existence of ‘multiple truths’ within our phenomenological approach; we reflected on how the qualitative research quality criteria of credibility, transferability, dependability and confirmability were considered in our methods; and we transparently described the limitations of our study. 

11. There are several minor grammatical and punctuation issues throughout that careful proof-reading may assist to identify.

After the adjustments based on the remarks of the editor and reviewers, the manuscript was thoroughly reviewed by one of the co-authors (SJ), a native English speakers (SJ). Oxford grammar and Fowler’s reference was used.

12. Table 1 has some additional, misplaced numbers (211, 212, 213, 214) that need removal.

The misplaced numbers are line numbers which have been automatically moved during the creation of the overall pdf in the submission system. In the original Microsoft Word file, these numbers are absent in the table. 

Reviewer #2: 

1. (lines 145, 454): The term interprofessional should be used rather than multidisciplinary. Multidisciplinary refers to activities performed by members of different academic disciplines. As previously defined, the term interprofessional should be used in this healthcare context. (see https://interprofessional.global/wp-content/uploads/2019/10/Guidance-on-Global-Interprofessional-Education-and-Collaborative-Practice-Research_Discussion-Paper_FINAL-WEB.pdf)

Thank you for your comment and for sharing this reference. Indeed, we didn’t mean ‘multidisciplinary’ but ‘interprofessional’. We have adjusted this in the methods and the results.

2. (line 205) The text of the results section, reads that 43% of the participants were employed in hospital care and 24% of the participants were employed in community care, however, this is not consistent with the results in Table 1 in the Healthcare settings section (48.6% and 27.0%, respectively).

Also, regarding that same section (Healthcare settings) in Table 1, the n-values total 313 rather than at least 340 interviews. What is the reason for this number being less than the total interviews? Did some people not respond to the question? Why would it not be 340 or greater particularly given that many of these healthcare professionals may be employed in more than one healthcare setting?

Thank you for identifying these inconsistencies. The numbers and corresponding percentages were not correct in both the text and the table and have been adjusted and double checked. Based on your comment, we have also checked the other numbers to be sure no errors or inconsistencies were reported. The total n for the variable ‘healthcare setting’ is indeed more than 340 (n= 352) and the total percentage 103.5% because of some respondents were employed in more than one setting.

3. (line 165-167) As interviews were conducted in the workplace or an alternative location, can you comment on participant body language, tone of voice, and/or other non-verbal cues that might be important indicators to the responses to the research questions?

Field notes were made during the interviews and included in the transcripts. During the process of transcribing, critical reflection and coding, these notes were taken into account, but, without specific overall analysis of these non-verbal cues. We acknowledge this limitation and have added the following text to the discussion: “Another limitation is the absence of structured integration of the field notes that have been made during the process of transcribing, critical reflecting and coding. Therefore, we might have missed important non-verbal indicators, such as participants’ body language and tone of voice.”

4. Did you collect information about cultural and/or ethnic diversity of the study participants? Did you consider that some participants might identify with a gender other than male or female? Could either or both cultural/ethnic background or gender identity affect perspectives related to interprofessional collaboration in pharmaceutical care?

We thought some participants might identify with a gender other than male or female. The answering options for ‘gender’ were: male, female, other. No respondents answered ‘other’. In the revised demographic table, we have added the category ‘other’ and indicated that no participants chose this option.

It is true that either or both cultural/ethnic background or gender identity could have affected perspectives related to interprofessional collaboration. However, we didn’t ask cultural or ethnic diversity in the short demographic questionnaire before the interview. This might have been an added value in the interpretation of the answers. We have acknowledged this in the limitations of the study as follows: “Socio-cultural influences, mainly in terms of attitudes towards other professions might affect perspectives related to interprofessional collaboration, as was demonstrated in several studies.(Irajpour, 2015; Schwappach, 2016) In this research, no information was sought on cultural and/or ethnic identities of respondents. We wished to avoid sensitive questions and any possibility that respondents might be identified by local readers. Diversity should be taken into account in future research.

5. Limitations:

(line 518) As the interviews were conducted by nurses, this could be a limitation as this may have potentially affected the responses from physicians, pharmacists, and other nurses. This limitation should be mentioned in the limitations section of the paper.

Thank you. By acknowledging this limitation in the discussion, we aimed to establish transparency. We have added the following paragraph to the discussion (limitations): “Researchers from all countries were trained in qualitative research, in-depth interviewing and ‘bracketing’ their own beliefs about nurses’ role in pharmaceutical care during a joint one-week training program. However, potential contextual intersecting relationships between the participants and the researchers cannot be ignored. As we wanted to avoid that the profession of the researchers would influence the responses from physicians, pharmacists, and other nurses, interviewers were asked not to inform interviewees about their profession if not questioned by the interviewee. By acknowledging this limitation in the discussion, we aimed to establish the transparency in our study.”

6. Overall, there are a few formatting errors such as in Table 1 the line numbers 211-214 are inside the cell.

Also, the writing could be further edited for clarity and conciseness for the reader.

The misplaced numbers are line numbers which have been automatically moved during the creation of the overall pdf in the submission system. In the original Microsoft Word file, the numbers are absent. 

After the adjustments based on the remarks of the editor and reviewers, the manuscript was thoroughly reviewed by co-author Professor Susan Jordan, a native English speaker.

---

## [Decision Letter · Decision Letter 1]

7 May 2021

Perspectives of nurses' role in interprofessional pharmaceutical care across 14 European countries: a qualitative study in pharmacists, physicians and nurses

PONE-D-20-36633R1

Dear Dr. De Baetselier,

We’re pleased to inform you that your manuscript has been judged scientifically suitable for publication and will be formally accepted for publication once it meets all outstanding technical requirements.

Kind regards,

Carl Richard Schneider, BN, BPharm (Hon), PhD

Academic Editor

PLOS ONE

Additional Editor Comments (optional):

Reviewers' comments:

Reviewer's Responses to Questions

**Comments to the Author**

1. If the authors have adequately addressed your comments raised in a previous round of review and you feel that this manuscript is now acceptable for publication, you may indicate that here to bypass the “Comments to the Author” section, enter your conflict of interest statement in the “Confidential to Editor” section, and submit your "Accept" recommendation.

Reviewer #1: All comments have been addressed

Reviewer #2: All comments have been addressed

2. Is the manuscript technically sound, and do the data support the conclusions?

Reviewer #1: Yes

Reviewer #2: (No Response)

3. Has the statistical analysis been performed appropriately and rigorously? 

Reviewer #1: Yes

Reviewer #2: (No Response)

4. Have the authors made all data underlying the findings in their manuscript fully available?

Reviewer #1: Yes

Reviewer #2: (No Response)

5. Is the manuscript presented in an intelligible fashion and written in standard English?

Reviewer #1: Yes

Reviewer #2: (No Response)

6. Review Comments to the Author

Reviewer #1: Spelling error on page 5, line 105. The word "strenghts' is incorrectly spelt. The correct spelling is "strengths". Please change this.

Reviewer #2: (No Response)

7. PLOS authors have the option to publish the peer review history of their article (what does this mean?). If published, this will include your full peer review and any attached files.

Reviewer #1: No

Reviewer #2: **Yes: **Kathleen M. MacMillan

---

## [Editor Report · Acceptance letter]

12 May 2021

PONE-D-20-36633R1 

Perspectives of nurses’ role in interprofessional pharmaceutical care across 14 European countries: a qualitative study in pharmacists, physicians and nurses 

Dear Dr. De Baetselier:

I'm pleased to inform you that your manuscript has been deemed suitable for publication in PLOS ONE. Congratulations! Your manuscript is now with our production department. 

Kind regards, 

on behalf of

Dr. Carl Richard Schneider 

Academic Editor

PLOS ONE